# Vitamin D Status and Immune Response in Hospitalized Patients with Moderate and Severe COVID-19

**DOI:** 10.3390/ph15030305

**Published:** 2022-03-02

**Authors:** Tatiana L. Karonova, Igor V. Kudryavtsev, Ksenia A. Golovatyuk, Arthur D. Aquino, Olga V. Kalinina, Alena T. Chernikova, Ekaterina K. Zaikova, Denis A. Lebedev, Ekaterina S. Bykova, Alexey S. Golovkin, Evgeny V. Shlyakhto

**Affiliations:** 1Almazov National Medical Research Centre, 197341 Saint-Petersburg, Russia; igorek1981@yandex.ru (I.V.K.); ksgolovatiuk@gmail.com (K.A.G.); akino97@bk.ru (A.D.A.); olgakalinina@mail.ru (O.V.K.); arabicaa@gmail.com (A.T.C.); catherine3452@yandex.ru (E.K.Z.); doctorlebedev@yandex.ru (D.A.L.); bykova160718@gmail.com (E.S.B.); golovkin_a@mail.ru (A.S.G.); shlyakhto_ev@almazovcentre.ru (E.V.S.); 2Institute of Experimental Medicine, 197376 Saint-Petersburg, Russia

**Keywords:** COVID-19, SARS-CoV-2, vitamin D, 25(OH)D, Th cell subsets, T-follicular helpers, immune response

## Abstract

A low 25-hydroxyvitamin D (25(OH)D) level is considered as an independent risk factor for COVID-19 severity. However, the association between vitamin D status and outcomes in COVID-19 is controversial. In the present study we investigate the association between the serum 25(OH)D level, immune response, and clinical disease course in patients with COVID-19. A total of 311 patients hospitalized with COVID-19 were enrolled. For patients with a vitamin D deficiency/insufficiency, the prevalence of severe COVID-19 was higher than in those with a normal 25(OH)D level (*p* < 0.001). The threshold of 25(OH)D level associated with mortality was 11.4 ng/mL (*p* = 0.003, ROC analysis). The frequency of CD3+CD4+ T helper (Th) cells was decreased in patients with 25(OH)D level ≤ 11.4 ng/mL, compared to healthy controls (HCs). There were no differences in the frequency of naive, central memory (CM), effector memory (EM), and terminally differentiated effector memory Th cells in patients with COVID-19 compared to HCs. The frequency of T-follicular helpers was decreased both in patients with 25(OH)D level > 11.4 ng/mL (*p* < 0.001) and 25(OH)D level ≤ 11.4 ng/mL (*p* = 0.003) compared to HCs. Patients with 25(OH)D level > 11.4 ng/mL had an increased frequency of Th2 CM (*p* = 0.010) and decreased Th17 CM (*p* < 0.001). While the frequency of Th2 EM was significantly increased, the frequency of Th17 EM was significantly decreased in both groups compared to HCs. Thus, 25(OH)D level is an independent risk factor for the disease severity and mortality in patients with COVID-19. We demonstrate that the serum 25(OH)D level ≤ 11.4 ng/mL is associated with the stimulation of Th2 and the downregulation of Th17 cell polarization of the adaptive immunity in patients with COVID-19.

## 1. Introduction

The COVID-19 pandemic caused by the SARS-CoV-2 respiratory virus is one of the world’s most pressing problems, and many studies are devoted to its prevention and treatment. This is a serious challenge for societies and a great threat to global health [1]. Risk factors associated with the severe course of COVID-19 are known, including age ≥ 60 years, cardiovascular diseases, obesity, diabetes, chronic kidney disease, chronic lung disease, smoking, cancer, solid organ or hematopoietic stem cell transplant patients [2]. In addition, numerous studies reported a relationship between low serum 25-hydroxyvitamin D (25(OH)D) levels and increased rates or the severity of various infections, including tuberculosis [3], human immunodeficiency virus infection [4,5], and influenza [6]. Recent studies demonstrated that a low 25(OH)D level is considered as an independent risk factor for COVID-19 severity [7,8,9,10,11]. Previously, we showed an association between severe vitamin D deficiency and outcomes in patients with COVID-19 [12]. The results of the meta-analysis that included several observational studies with data for approximately two million adults suggested that vitamin D deficiency/insufficiency increased the susceptibility to COVID-19 and its severity. Moreover, some clinical benefits and an improvement in inflammatory markers were detected after vitamin D supplementation in the treatment of COVID-19 [13].

Vitamin D is a fat-soluble vitamin involved in maintaining the calcium-phosphorus metabolism [14,15]. On the other hand, many tissues and cells, including the cells of the immune system, have specific vitamin D receptors [16,17]. Vitamin D enhances innate cellular immunity by stimulating LL-37, an antimicrobial peptide residing in monocytes, B cells, natural killers (NK cells), epithelial cells, γδ T cells [18], and β-defensins, which activate chemotaxis and prevent viral infection [19]. Furthermore, vitamin D activates monocyte differentiation into macrophages and triggers autophagy, thus preventing severe immunopathology associated with viral infections [20]. Normal 25(OH)D levels can promote immunoregulatory functions in viral respiratory infection conditions and, overall, can influence the altered immune-inflammatory COVID-19 reactivity by downregulating overly abundant cytokine responses that comprise the pathological cytokine storm [21]. Vitamin D suppresses T-cell proliferation by inhibiting the secretion of interleukin -12 (IL-12) and leads to a shift from T helpers 1 (Th1) to a Th2 phenotype [22]. It also affects the maturation of T cells with a decrease in the inflammatory Th17 phenotype and an increase in T regulatory cells, resulting in the regulation of the production of inflammatory cytokines IL-17 and IL-21 with an increase in the production of the anti-inflammatory cytokine IL-10 [23]. In addition, there are a lot of pleiotropic effects of vitamin D on obesity, diabetes mellitus, and cardiovascular diseases that are known aggravating factors for COVID-19 [24].

Hence, the present study aims to (1) establish whether a low 25(OH)D level is a risk factor in the clinical course of COVID-19 and (2) investigate immune response markers in hospitalized patients with COVID-19 with different vitamin D statuses.

## 2. Results

### 2.1. Comparison According to Disease Severity

We analyzed data from 311 patients admitted to the hospital due to COVID-19. A total of 56% had moderate disease severity and 41% had severe disease. Patients were divided into groups according to COVID-19 severity. The results are presented in Table 1. Patients with mild disease were excluded from the analysis due to the small number of cases (*n* = 10 patients). Patients with severe COVID-19 were older than the patients with moderate disease—66 [58;75] and 59 [51;68] years, respectively, *p* < 0.001. In the group with severe disease, there was a higher prevalence of chronic conditions, such as obesity, diabetes mellitus (DM), arterial hypertension (AH), ischemic heart disease (IHD), and chronic kidney disease (CKD) (all *p* < 0.05). The group with moderate severity had lower levels of the C-reactive protein (CRP), ferritin, lactate dehydrogenase (LDH), glucose, and D-dimer, compared to the group with severe disease (all *p* < 0.05). The serum 25(OH)D level was lower in patients with severe disease compared to patients with moderate disease—14.8 [8.3;21.3] ng/mL and 18.9 [10.2;33.3] ng/mL, respectively, *p* = 0.001. Lymphocytes were significantly lower and the neutrophils/lymphocytes ratio (NLR) was significantly higher in the group with severe disease (*p* < 0.05).

### 2.2. Comparison According to Vitamin D Status

Patients were also divided into groups based on their vitamin D status at the time of hospitalization. A total of 69 patients had normal 25(OH)D levels, 185 had insufficiency, and 57 had deficiency (Table 2). Patients with a vitamin D deficiency/insufficiency were older compared to those with a normal vitamin D status—64 [55; 72] and 58 [47.5; 68.5], respectively, *p* = 0.002. There were no differences in the prevalence of obesity, DM, AH, and CKD between the groups. Patients with a vitamin D deficiency/insufficiency had a higher prevalence of severe COVID-19 than those with normal 25(OH)D levels—50.4% and 8.7%, *p* < 0.001. The percentage of lung involvement on the computer tomography (CT) was also higher in patients with a vitamin D deficiency or insufficiency, as well as the prevalence of intensive care unit (ICU) admission (*p* < 0.05).

Patients with COVID-19 with a vitamin D deficiency/insufficiency had higher levels of CRP, ferritin, LDH, glucose, and D-dimer at the time of hospitalization, compared to the vitamin D sufficient group (all *p* < 0.05). Thus, the baseline CRP was 79.7 [32.5; 146.7] mg/L in patients with a vitamin D deficiency/insufficiency and 28.3 [7.6; 62.5] mg/L in patients with a normal vitamin D status, *p* < 0.001. Lymphocytes were significantly lower and NLR significantly higher in those with a vitamin D deficiency/insufficiency (*p* < 0.05). There was a positive association between CRP and the percentage of lung involvement (r = 0.459, *p* < 0.001). Furthermore, LDH was positively associated with the percentage of lung involvement (r = 0.469, *p* < 0.001) (Table 2).

When comparing the data separately for patients with a normal level, insufficiency, and deficiency of vitamin D, there were also significant differences in NLR, CRP, ferritin, LDH, glucose, and D-dimer levels, the percentage of CT involvement, and the duration of hospitalization (all *p* < 0.05). The results are shown in Table 3.

To assess the predictive value of 25(OH)D levels and find the optimal classification threshold, a ROC analysis was performed. The threshold for 25(OH)D levels associated with mortality was 11.4 ng/mL (AUC area = 0.811; sensitivity, 76%; and specificity, 77%; *p* = 0.003) (Figure 1a) and the threshold for 25(OH)D levels associated with severe disease was 11.7 ng/mL (AUC area = 0.69; sensitivity, 76%; and specificity, 54%; *p* = 0.01) (Figure 1b).

Multiple logistic regression analysis identified significant risk factors for severe COVID-19: age > 60 years (OR 1.38; 95% CI = 1.18–1.74; *p* = 0.02), obesity (OR 2.34; 95% CI = 1.56–3.71; *p* = 0.003), and serum 25(OH)D levels < 11.4 ng/mL (OR 1.29; 95% CI = 1.21–1.38; *p* = 0.01). The male gender was not a significant risk factor for severe disease as well as the presence of diabetes and arterial hypertension.

### 2.3. Immune Status in Patients with COVID-19 and Vitamin D Deficiency and Insufficiency

#### 2.3.1. Decreased Frequency of Th Cells According to Vitamin D Status

We analyzed immune markers, including the T-cell population, in 79 randomly selected patients with COVID-19 and 28 healthy controls (HCs). According to the 25(OH)D threshold level of 11.4 ng/mL associated with mortality in this study, 79 patients with COVID-19 were divided into 2 groups: the first group consisted of patients with a serum 25(OH)D level ≤ 11.4 ng/mL (*n* = 20)), the second group with a serum 25(OH)D level > 11.4 ng/mL (*n* = 59). The study design is provided in Figure 2.

To study the possible alterations of CD3+CD4+ Th cell subsets, we first determined the relative and absolute numbers of circulating CD3+CD4+ Th cells in the peripheral blood of both patient groups and HCs (Figure 3a,b). The frequency of Th cells in patients with COVID-19 and a serum 25(OH)D level ≤ 11.4 ng/mL was significantly lower than in HC (33.22% [27.59; 42.77] vs. 41.64% [36.67; 47.22], *p* = 0.038). However, both groups of patients with COVID-19 exhibited reduced absolute numbers of Th cells (400 cells/μL [255; 593] and 391 cells/μL [286; 626] in patients with 25(OH)D ≤ 11.4 ng/mL and 25(OH)D > 11.4 ng/mL, respectively, vs. 813 cells/μL [683; 903] in HC with *p* < 0.001 in both cases.

#### 2.3.2. Alterations in the Main T-Cell Subsets in Patients with COVID-19 According to Vitamin D Status

Next, we addressed the analysis of differentiation of Th subsets according to CD45RA and CD62L expression and distinguished four main stages of Th maturation—“naïve” (CD45RA+CD62L+), central memory (CM) (CD45RA−CD62L+), effector memory (EM) (CD45RA−CD62L–), and T effector memory re-expressing CD45RA (TEMRA) (CD45RA+CD62L–) cells. Regarding the percentage of circulating Th subsets at different maturation checkpoints, no significant differences were found between all three groups. However, we noticed that the frequencies of all four Th subsets at different stages of maturation in patients with COVID-19 with 25(OH)D > 11.4 ng/mL were significantly lower than those in the control group (Figure 4e–h). Similarly, patients with COVID-19 with 25(OH)D ≤ 11.4 ng/mL also demonstrated significantly lower absolute counts of main Th cell subsets when compared with HC, except for TEMRA Th cells.

#### 2.3.3. Imbalance of the Central Memory “Polarized” Th Cell Subsets According to Vitamin D Status

To identify the circulating “polarized” Th cell, we screened the expression of four chemokine receptors—CXCR5, CXCR3, CCR6, and CCR4—on circulating central memory and effector memory Th cells. Flow cytometry analysis revealed significantly higher proportion of Th2 (15.55% [12.46; 19.89] vs. 11.98% [9.87; 16.69], *p* = 0.010) and significantly reduced proportions of Th17 (32.60% [26.79; 36.22] vs. 38.73% [34.12; 44.93], *p* < 0.001) and follicular Th (Tfh) (13.54% [10.91; 17.18] vs. 17.82% [15.56; 20.77], *p* < 0.001) cells in peripheral blood samples from patients with COVID-19 with 25(OH)D > 11.4 ng/mL compared to HCs (Figure 5). Similarly, patients with 25(OH)D ≤ 11.4 ng/mL demonstrated the decreased frequency of Tfh cells compared to HCs (13.70% [10.53; 14.83] vs. 17.82% [15.56; 20.77], *p* = 0.003).

#### 2.3.4. Alterations in the Tfh Subsets in Patients with COVID-19 According to Vitamin D Status

Next, we analyzed the CXCR5+ CM Tfh cell populations in patients with COVID-19 and in the control group. Based on the cell-surface expression of chemokine receptors CXCR3 and CCR6, we identified CXCR3+CCR6− Tfh1, CXCR3−CCR6− Tfh2, and CXCR3−CCR6+ Tfh17 cells, as was suggested by Morita and co-authors [25]. Our overall analysis of these different Tfh cell subsets indicated that only patients with 25(OH)D > 11.4 ng/mL showed significant differences with HC; we found a significantly higher proportion of Tfh2 (22.21% [16.46; 26.65] vs. 16.59% [14.68; 21.28], *p* = 0.006) and Tfh17 (32.88% [24.24; 41.39] vs. 28.01% [19.88; 33.70], *p* = 0.029) and significantly reduced proportions of Tfh1 (28.14% [22.69; 34.37] vs. 32.25% [26.79; 41.40], *p* = 0.018) and double-positive (DP) Tfh (13.84% [10.82; 18.56] vs. 19.96% [17.67; 21.71], *p* < 0.001) cells in peripheral blood samples compared to HCs (Figure 6). Interestingly, we observed the absence of alterations in Tfh cell subset distribution in patients with 25(OH)D ≤ 11.4 ng/mL, when compared to HCs.

#### 2.3.5. Imbalance of Effector Memory “Polarized” Th Cell Subsets According to Vitamin D Status

Finally, we identified circulating “polarized” Th cell subsets within the total effector memory CD45RA−CD62L– Th cell subset that could leave the circulating blood and migrate to diverse sites of inflammation to promote pathogen clearance. Similar to CM Th cells, we revealed a significantly higher frequency of Th2 (2.85% [1.96; 4.19] vs. 1.30% [0.87; 1.63], *p* < 0.001) and significantly reduced numbers of Th17 (43.25% [33.72; 52.45] vs. 54.43% [44.88; 66.52], *p* = 0.001) cells in peripheral blood samples from patients with 25(OH)D > 11.4 ng/mL, when compared to HCs (Figure 7). Furthermore, patients with 25(OH)D ≤ 11.4 ng/mL also exhibited higher levels of Th2 (1.92% [1.49; 3.95] vs. 1.30% [0.87; 1.63], *p* < 0.001) and lower levels of Th17 (37.11% [33.16; 45.45] vs. 54.43% [44.88; 66.52], *p* < 0.001), when compared to HCs, but we also found the decreased frequency of circulating Tfh cells in this group when compared to HCs (5.01% [3.62; 9.87] vs. 10.06% [7.27; 14.16], *p* = 0.016).

## 3. Discussion

The results of previous studies have shown a high prevalence of vitamin D insufficiency or deficiency in patients hospitalized with COVID-19 [7,8,9,10]. In the present study, we also found that most patients with COVID-19 had low serum 25(OH)D levels. Moreover, vitamin D deficiency/insufficiency was associated with a higher prevalence of severe COVID-19. These observations have been previously confirmed by the meta-analysis of I. Chiodini et al. included 54 studies in their research and demonstrated an association of severe vitamin D deficiency or insufficiency with a high risk of SARS-CoV-2 infection, COVID-19-related hospitalization, COVID-19-related ICU admissions, and COVID-19-related mortality [26]. Similarly, H. AlSafar et al. assessed serum 25(OH)D levels in 464 hospitalized patients with COVID-19. They found an association between 25(OH)D < 12 ng/mL and a 2.58-fold (95% CI, 1.01, 6.62) increased risk of COVID-19 mortality following an adjustment for age, comorbidities, or sex (*p* = 0.048) [27].

It is known that the severity of COVID-19 is due to the manifestations of a cytokine storm, which is accompanied by the dysregulation of both innate and adaptive immunity [28,29,30]. However, the relationship between vitamin D status and immune markers in patients with COVID-19 is unclear. However, there are several discussions in the literature concerning the use of vitamin D supplementation in COVID-19 patients as a possible way to decrease the cytokine storm and improve the prognosis. One study from Italy showed that patients with acute COVID-19 had the lowest serum 25(OH)D levels; additionally, they manifested high-circulating IL-6 at admission, which decreased after calcitriol administration [31].

Vitamin D immunological activities affect components of the innate and adaptive immune system and regulate immune cell subset differentiation and maturation, antigen processing and presentation, and the production of different cytokines and chemokines. The protective effect of vitamin D is closely linked with its capacity to negatively regulate the differentiation and maturation of dendritic cells and, especially, to downregulate the expression of co-stimulatory molecules CD80 and CD86 as well as MHC class II molecules on the cell surface of dendritic cells [32], and to decrease IL-12 and IL-23 production [33]. Furthermore, vitamin D has been found to suppress the adaptive immune response by the modulation of antigen-presentation and T-cell proliferation and “polarization”, thus resulting in the downregulation of inflammation and immune tolerance promotion [34]. It was shown that the effect of vitamin D was mostly mediated by IL-4 and results in augmenting Th2 development, Th1 inhibiting, and in the increased expression of the transcription factor GATA-3 [22].

Moreover, the activated form of vitamin D (calcitriol) appeared to be involved in two independent mechanisms of “pro-inflammatory” Th17 response suppression, including the inhibition of IL-17 production by CD4+ T cells and the reduction in the ability of dendritic cells to support the differentiation of Th0 cells toward Th17 [35,36]. Furthermore, vitamin D deficiency induced the differentiation and development of Th17 cells [37]. Additionally, vitamin D was able to promote the differentiation of regulatory T cells that prevent and decrease inflammatory responses by inducing the anti-inflammatory cytokine IL-10 and the FoxP3 transcription factor [38]. It was shown that vitamin D was able to affect the inflammatory response by modulating the migration and homing properties of Th cells by upregulating the skin-homing receptor, CCR10 expression, and inhibiting the gut-homing receptors alpha4beta7 and CCR9 CXCR3 [39]. Moreover, in vitro migration of memory CCR6+ Th cells toward CCL20, the ligand for CCR6, was significantly decreased after treatment with 1,25(OH)2D3, preventing Th17 cells from migrating to the sites of inflammation [40].

Circulating CD4+ T cells possess a range of helper and effector functions, and they are important for the control of different types of innate and adaptive immune effector mechanisms, as well as the clearance of almost all viral infections, including SARS-CoV-2 [41]. Furthermore, the prompt and effective induction of SARS-CoV-2-specific CD4+ T cells and their presence in circulation within 2–4 days post-symptom onset in acute COVID-19 were associated with mild/moderate COVID-19 and good prognosis [42,43]. Controversially, the rapid induction of humoral response and the prolonged absence of SARS-CoV-2-specific CD4+ T cells were linked with an increase in disease severity and poor COVID-19 outcomes [43,44]. Thus, we can speculate that dramatic changes in Th cell subset proportions could reflect the presence of effective immune response linked with the control and resolution of SARS-CoV-2 infection. Controversially, the absence of any reported alterations in Th cell subsets, except Tfh cells, could reflect the ineffective immunity to infection and was strongly associated with high severity and a risk of fatal COVID-19.

## 4. Materials and Methods

### 4.1. Patients

The present study included 311 patients from the Almazov National Medical Research Centre (St. Petersburg, Russia), hospitalized with COVID-19 (150 women, 161 men). All patients signed informed consent for participation. COVID-19 was confirmed by the polymerase chain reaction (PCR) test and/or CT scan. The study was approved by the local Ethics Committee of Almazov National Medical Research Centre (protocol No. 1011-20-02C, 30 November 2020) and complied with the Helsinki Declaration. This was a single-center open-label study performed during the period ranging from 30 November 2020 to 20 March 2021. The inclusion criteria were the following: (i) 18 to 75 years old, (ii) a positive PCR test for SARS-CoV-2, and (iii) signed informed consent. We did not include subjects who had a daily vitamin D intake of more than 1000 IU. Additionally, the exclusion criteria were primary hyperparathyroidism or hypercalcemia of other etiologies; clinically significant gastrointestinal diseases ot kidneys pathology (estimated Glomerular Filtration Rate (eGRF) less than 45 mL/min/1.73 m^2^); liver diseases, which can influence vitamin D absorption and metabolism; a history of granulomatous diseases; oncology diseases in their medical history (less than 5 years); and alcohol and drug addiction. Pregnant or breastfeeding women did not participate in the study. If potential participants had other circumstances that the investigator considered as inappropriate, they were not allowed to participate in the study.

A total of 79 patients with COVID-19 showing a 25(OH)D level ≤ 11.4 ng/mL (*n* = 20, aged 58 years (53;75), 12 men and 8 women) and 25(OH)D level > 11.4 ng/mL (*n* = 59, age 58 (52;67), 25 men and 34 women), and 28 apparently healthy donors (aged 40 years (35;62), 17 men and 11 women) were enrolled in the present study aimed to investigate circulating T-cell subsets using flow cytometry.

Baseline characteristics were recorded for all patients, including medical history, height, weight, and body mass index (BMI). CT data were also evaluated for each patient. We used chest CT without intravenous contrast enhancement to detect pneumonia. The volume of lung tissue lesions was described as follows: CT-1, lesion volume < 25%; CT-2, lesion volume 25–50%; CT-3, lesion volume 50–75%; and CT-4, lesion volume > 75% [45]. After signing the informed consent and baseline examination, patients were divided into groups according to COVID-19 severity. According to the guidelines [45], the severity was determined by the following criteria: mild illness—temperature < 38 °C, absence of breath shortness, dyspnea or normal CT; moderate illness—temperature > 38 °C, SpO2 < 95%, CRP > 10 mg/L, CT—1 or 2; and severe illness—hemodynamic instability, SpO2 < 93%, CT—3 or 4. We also assessed ICU hospitalizations, the duration of hospitalization, and the outcome of the disease.

### 4.2. Physical Data

The anthropometric examination included the height (cm) and weight (kg), based on which the BMI was calculated (in kg/m^2^).

### 4.3. Laboratory Tests

Serum 25(OH)D level was detected by chemiluminescence immunoassay on microparticles (Abbott Architect i8000, Chicago, IL, U.S.A.), intra-assay CV of 1.60–5.92%; the inter-assay CV ranged from 2.15 to 2.63%. Normal vitamin D status was considered to be 25(OH)D ≥30 ng/mL (≥75 nmol/L); for insufficiency, ≥20 and <30 ng/mL (≥50 and <75 nmol/L); and for deficiency, <20 ng/mL (<50 nmol/L) [46]. The reference interval for the serum 25(OH)D level was 3.4–155.9 ng/mL.

Additionally, we evaluated the biochemical parameters, such as fasting plasma glucose (FPG) and creatinine levels (Roche Diagnostics GmbH, Mannheim, Germany).

We also assessed the acute phase proteins CRP, LDH, and ferritin. An automatic biochemistry analyzer (Cobas Integra 400 (Roche Diagnostics GmbH, Mannheim, Germany)) and corresponding diagnostic kits from the same manufacturer were used to determine the level of CRP by means of the turbidimetric method (reference range, 0–5 mg/L). The ferritin level was measured on an Abbott Architect c8000 analyzer (Chicago, IL, USA.; reference range, 64–111 nmol/L). The D-dimer was assessed by an automatic analyzer (ACL TOP 300 CTS, (Bedford, MA, USA)); reference range 0–0.55 µg/mL FEU.

All investigations of the T-cell subsets were performed less than 6 h after blood collection. Peripheral blood samples were collected into vacuum test tubes containing K_3_-EDTA anticoagulant and were then processed to analyze the relative and absolute numbers of the main T- and B-cell subsets by multicolor flow cytometry. Clinical blood analysis was performed using a Cell-DYN Ruby-Hematology Analyzer (Abbott, Abbot Park, Chicago, IL, USA). T-cell immunophenotyping was performed by multicolor flow cytometry using a CytoFlex S flow cytometer (Beckman Coulter, Indianapolis, IN, USA). The flow cytometry data was analyzed using Kaluza software v2.1 (Beckman Coulter, Inc., Indianapolis, IN, USA).

The staining procedure, gating strategy, and analysis of the results were performed as described earlier [27]. Optimal combinations of the antibodies conjugated with various fluorochromes were used according to a previously published method. This method was developed by Sallusto et al. [47] and validated by our group [48,49].

Briefly, a whole peripheral blood (100 μL) sample was stained using FITC-labeled mouse anti-human CD45RA (clone ALB11, cat. IM0584U, Beckman Coulter, Indianapolis, IN, USA); PE-labeled mouse anti-human CD62L (clone DREG56, cat. IM2214U, Beckman Coulter, Indianapolis, IN, USA); PerCP/Cy5.5-labeled mouse anti-human CXCR5 (CD185, clone J252D4, cat. 356910, BioLegend, Inc., San Diego, CA, USA); PE/Cy7-labeled mouse anti-human CCR6 (CD196, clone G034E3, cat 353418, BioLegend, Inc., San Diego, CA, USA); APC-labeled mouse anti-human CXCR3 (CD183, clone G025H7, cat. 353708, BioLegend, Inc., San Diego, CA, USA); APC-Alexa Fluor 750-labeled mouse anti-human CD3 (clone UCHT1, cat. A94680, Beckman Coulter, Indianapolis, IN, USA); Pacific Blue-labeled mouse anti-human CD4 (clone 13B8.2, cat. B49197, Beckman Coulter, Indianapolis, IN, USA); and Brilliant Violet 510-labeled mouse anti-human CCR4 (CD194, clone L291H4, cat. 359416, BioLegend, Inc., San Diego, CA, USA). The staining protocols were performed according to the manufacturer’s recommendations. Samples were stained with the antibodies mentioned above at room temperature for 15 min in the dark. Next, the erythrocytes were lysed by adding 1 mL of VersaLyse Lysing Solution (Beckman Coulter, Inc., Indianapolis, IN, USA) with 25 μL of IOTest 3 Fixative Solution (Beckman Coulter, Inc., Indianapolis, IN, USA) in a dark at room temperature for 15 min. Next, all the samples were washed (330× *g* for 8 min) twice with sterile PBS supplemented with 2% of fetal calf serum (FCS) (Sigma-Aldrich Co., Saint Louis, MO, USA), resuspended in 500 μL of fresh PBS with 2% neutral formalin (cat. HT5011-1CS, Sigma-Aldrich Co., Saint Louis, MO, USA), and subjected to flow cytometry analysis. At least 40,000 CD3+CD4+ Th cells were collected from each sample.

Blood samples for 25(OH)D assessment were taken in the morning from the cubital vein, centrifuged, aliquoted, and stored in a freezer at a temperature of −70 °C before testing was performed.

### 4.4. Statistical Analysis

Statistical processing of the research results was carried out using the IBM SPSS Statistics for Windows ver. 26 (IBM Corp., Armonk, NY, USA), with the help of standard methods of variation statistics. The between-group comparison was carried out using the Mann–Whitney criteria for non-normal distribution; the results are presented as the median (Me) and interquartile range [25%; 75%] or as the mean (M) and standard deviation (SD) for the Student criterion in normally distributed parameters. Associations between the quantitative parameters were assessed using Spearman’s correlation coefficient. The search for the cutoff of the predictive factors was based on the receiver operating characteristic (ROC analysis) construction with the calculation of sensitivity and specificity. The optimal cutoff was the level corresponding to the maximum sum of sensitivity and specificity obtained in the ROC analysis. We explored the association between 25(OH)D level and COVID-19 severity using logistic regression, with results expressed as the β coefficients and 95% CI. The criterion for the statistical reliability of the obtained results was *p* < 0.05.

## 5. Conclusions

Our study reveals that most hospitalized patients with COVID-19 have a vitamin D insufficiency or deficiency, which is associated with severe COVID-19, and 25(OH)D level ≤ 11.4 ng/mL could be associated with an increased risk of mortality. We found a higher frequency of Th2 and reduced frequency of Th17 and Tfh cells in peripheral blood samples from patients with COVID-19 and a serum 25(OH)D level > 11.4 ng/mL compared to HCs. Additionally, in patients with COVID-19 and a serum 25(OH)D level ≤ 11.4 ng/mL, only a decrease in the frequency of Tfh cells was observed. The analysis of CM Tfh cell subsets indicated that only patients with a serum 25(OH)D level > 11.4 ng/mL showed significant differences with HCs. The obtained results could serve as the basis for future research on the role of vitamin D deficiency and its effects on immune markers and disease progression.

## Figures and Tables

**Figure 1 pharmaceuticals-15-00305-f001:**
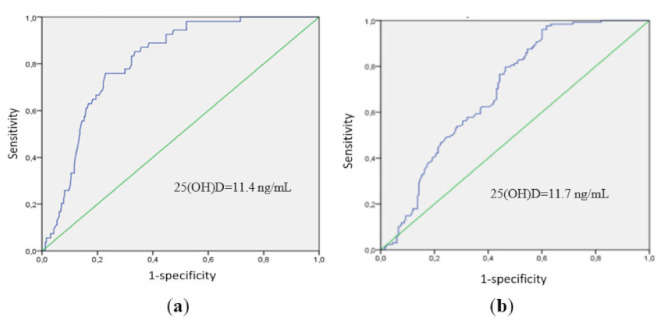
ROC analysis data: (**a**) the threshold for 25(OH)D levels associated with mortality was 11.4 ng/mL (AUC area = 0.811; sensitivity, 76%; and specificity, 77%; *p* = 0.003), and (**b**) the threshold for 25(OH)D levels associated with severe disease was 11.7 ng/mL (AUC area = 0.69; sensitivity, 76%; and specificity, 54%; *p* = 0.01).

**Figure 2 pharmaceuticals-15-00305-f002:**
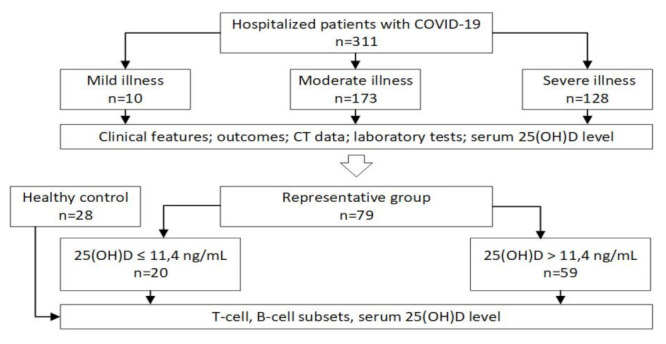
Study design: 25(OH)D—25-hydroxyvitamin D; CD—cluster of differentiation.

**Figure 3 pharmaceuticals-15-00305-f003:**
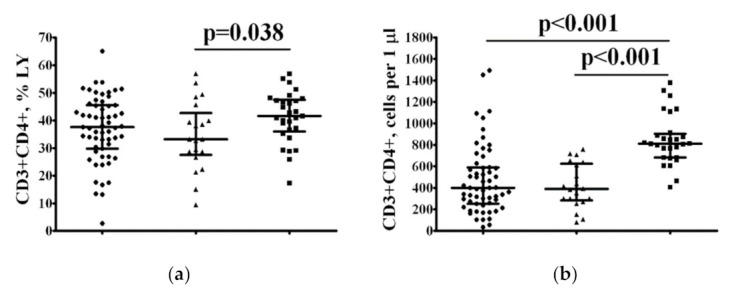
The frequency of peripheral blood CD3+CD4+ Th cells in patients with COVID-19 and 25(OH)D > 11.4 ng/mL or 25(OH)D ≤ 11.4 ng/mL. Scatter plots show the relative (**a**) and absolute (**b**) numbers of CD3+CD4+ T-lymphocytes in peripheral blood samples from patients with COVID-19 and 25(OH)D > 11.4 ng/mL (black circles, *n* = 59), patients with COVID-19 and 25(OH)D ≤ 11.4 ng/mL (black triangles, *n* = 20), and HC subjects (*n* = 28, black squares). Numbers represent the percentages of the indicated Th cell subset among the total lymphocyte population. Each dot represents an individual subject, and the horizontal bars represent the group medians and interquartile ranges (Me [IQR 25; 75]).

**Figure 4 pharmaceuticals-15-00305-f004:**
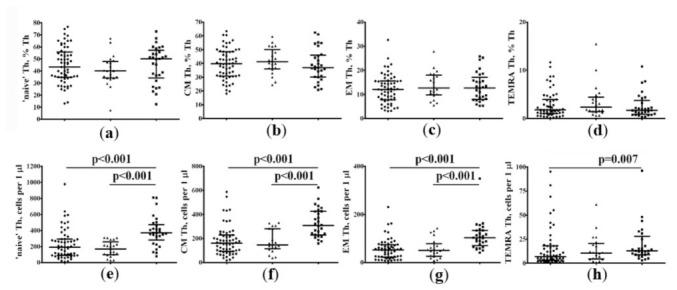
Distribution of differentiation Th subsets according to CD45RA and CD62L expression in patients with COVID-19 and in HCs. Scatter plots showing the relative (**a**–**d**) and absolute (**e**–**h**) counts of the main Th cell maturation stages in peripheral blood samples from patients with COVID-19 with 25(OH)D > 11.4 ng/mL (black circles), patients with COVID-19 with 25(OH)D≤ 11.4 ng/mL (black triangles) and HC subjects (black squares). Scatter plots (**a**,**e**)—”naïve” CD45RA+CD62L+ Th cells; (**b**,**f**)—central memory CD45RA−CD62L+ Th cells; (**c**,**g**)—effector memory CD45RA−CD62L– Th cells; and (**d**,**h**)—TEMRA CD45RA+CD62L– Th cells. Numbers represent the percentage of the indicated Th cell subset among the total lymphocyte population. Each dot represents the individual subjects, and the horizontal bars represent the group medians and quartile ranges (Me [IQR 25; 75]).

**Figure 5 pharmaceuticals-15-00305-f005:**
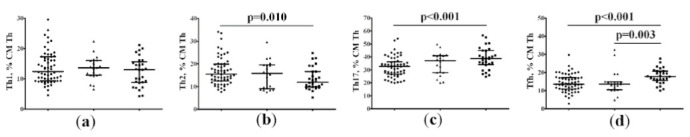
Imbalance of central memory CD3+CD4+ Th cell subsets in patients with COVID-19. Scatter plots show the relative count of the main “polarizes” Th cell subsets within the total central memory CD45RA−CD62L+ Th cells, including Th1 (CXCR5-CXCR3+CCR6-CCR4-) (**a**), Th2 (CXCR5-CXCR3-CCR6-CCR4+) (**b**), total Th17 (CXCR5-CCR6+) (**c**), and CXCR5-positive Tfh cells (**d**), respectively, in peripheral blood samples from patients with 25(OH)D > 11.4 ng/mL (black circles), patients with 25(OH)D ≤ 11.4 ng/mL (black triangles), and HC subjects (black squares). Numbers represent the percentage of the indicated Th cell subset among the total lymphocyte population. Each dot represents the individual subjects, and the horizontal bars represent the group medians and quartile ranges (Me [IQR 25; 75]).

**Figure 6 pharmaceuticals-15-00305-f006:**
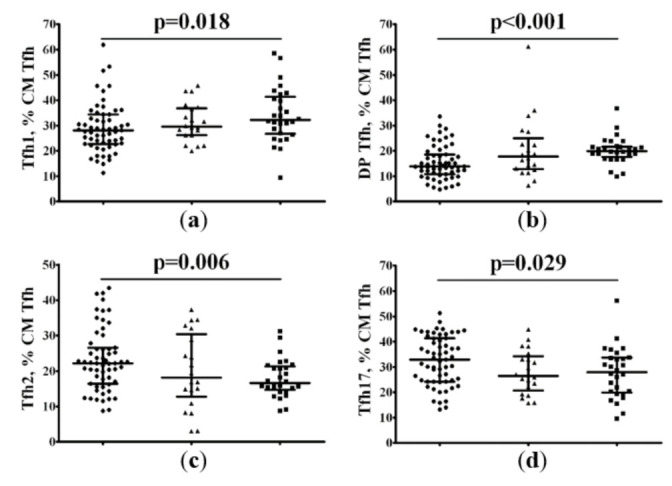
Imbalance of central memory Tfh cell subsets in patients with COVID-19. Scatter plots show the percentages of CXCR3+CCR6− Tfh1 (**a**), double-positive CXCR3+CCR6+ Tfh (**b**), CXCR3−CCR6− Tfh2 (**c**), and CXCR3−CCR6+ Tfh17 (**d**) cells among the total CD45RA−CD62L+ Tfh population, respectively, in peripheral blood samples in patients with 25(OH)D > 11.4 ng/mL (black circles), patients with 25(OH)D ≤ 11.4 ng/mL (black triangles), and HC subjects (black squares). Numbers represent the percentage of the indicated Th cell subset among the total lymphocyte population.

**Figure 7 pharmaceuticals-15-00305-f007:**
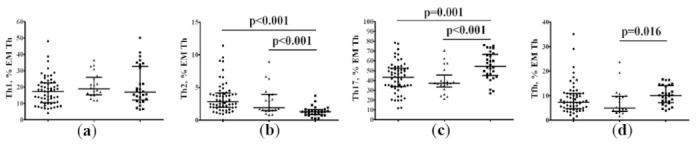
Imbalance of effector memory CD3+CD4+ Th cell subsets in patients with COVID-19. Scatter plots showing the relative count of the main “polarizes” Th cell subsets within the total central memory CD45RA−CD62L– Th cells, including Th1 (CXCR5-CXCR3+CCR6-CCR4-) (**a**), Th2 (CXCR5-CXCR3-CCR6-CCR4+) (**b**), total Th17 (CXCR5-CCR6+) (**c**), and CXCR5-positive Tfh cells (**d**), respectively, in peripheral blood samples from patients with 25(OH)D > 11.4 ng/mL (black circles), patients with 25(OH)D ≤ 11.4 ng/mL (black triangles), and HC subjects (black squares). Numbers represent the percentage of the indicated Th cell subset among the total lymphocyte population.

**Table 1 pharmaceuticals-15-00305-t001:** Patients’ baseline characteristics depending on COVID-19 severity.

Parameters	Moderate (*n* = 173)	Severe (*n* = 128)	*p*
Age, years, Me + IQR [25; 75]	59 [51; 68]	66 [58; 75]	<0.001
Gender, female, *n* (%)	89 (51.4)	61 (47.7)	0.52
Days from symptom onset to hospitalization, days, Me + IQR [25; 75]	8 [6; 11]	8 [6; 11]	0.92
Bed days, days, Me + IQR [25; 75]	15 [13; 20]	19 [13; 27]	0.003
CT lung involvement, %, Me + IQR [25; 75]	30 [20; 45]	68 [55; 76]	<0.001
BMI, kg/m^2^, Me + IQR [25; 75]	29 [26; 32]	29 [25; 34]	0.66
Obesity, *n* (%)	69 (39.9)	55 (43.0)	0.048
DM type 2, *n* (%)	47 (27.2)	55 (43.0)	0.004
AH, *n* (%)	123 (71.1)	111 (86.7)	0.001
IHD, *n* (%)	37 (21.4)	66 (51.6)	<0.001
CKD, *n* (%)	13 (7.5)	21 (16.4)	0.017
Neutrophils, ×10^9^/L, Me + IQR [25; 75]	4.8 [3.4; 7.3]	5.7 [3.4; 8.1]	0.16
Lymphocytes, ×10^9^/L, Me + IQR [25; 75]	1.2 [0.8; 1.5]	0.9 [0.7; 1.3]	0.002
NLR, Me + IQR [25; 75]	4.2 [2.7; 7.1]	5.7 [2.9; 10.1]	0.007
25(OH)D, ng/mL, Me + IQR [25; 75]	18.9 [10.2; 33.3]	14.8 [8.3; 21.3]	0.001
Plasma glucose, mmol/L, Me + IQR [25; 75]	6.5 [5.9; 8.0]	7.7 [6.5; 9.3]	<0.001
CRP, mg/L, Me + IQR [25; 75]	39.6 [20.7; 91.4]	93.1 [53.4; 190.7]	<0.001
Ferritin, ng/mL, Me + IQR [25; 75]	488 [234; 825]	776 [381; 1296]	<0.001
LDH, u/L, Me + IQR [25; 75]	341 [267; 482]	541 [399; 726]	<0.001
D-dimer, µg/mL FEU, Me + IQR [25; 75]	0.3 [0.2; 0.5]	0.6 [0.4; 0.9]	<0.001
Creatinine, mcmol/L, Me + IQR [25; 75]	68 [58; 77]	68 [51; 83]	0.74

CT—computer tomography; BMI—body mass index; DM—diabetes mellitus; AH—arterial hypertension; IHD—ischemic heart disease; CKD—chronic kidney disease; NLR—neutrophils/lymphocytes ratio; 25(OH)D—25-hydroxyvitamin D; CRP—C-reactive protein; LDH—lactate dehydrogenase; Me—median; and IQR—interquartile range.

**Table 2 pharmaceuticals-15-00305-t002:** Patients’ baseline characteristics depending on vitamin D status.

Parameters	Vitamin D Status		*p*
	Normal (*n* = 69)	Deficiency/Insufficiency (*n* = 242)	
Age, years, Me + IQR [25; 75]	58 [48; 69]	64 [55; 72]	0.002
Gender, female, *n* (%)	37 (53.6)	120 (49.6)	0.550
Disease severity, *n* (%)			
Mild Moderate Severe	7 (10.2) 56 (81.1) 6 (8.7)	3 (1.3) 117 (48.3) 122 (50.4)	< 0.001
Vitamin D status, *n* (%)			
Normal Insufficiency Deficiency Severe deficiency	69 (100) - - -	- 58 (23.9) 184 (76) 102 (42.1)	
Days from symptom onset to hospitalization, days, Me + IQR [25; 75]	7 [4; 10]	8 [6; 11]	0.050
Bed days, days, Me + IQR [25; 75]	14 [9; 19]	17 [13; 23]	0.002
ICU, *n* (%)	9 (13)	97 (40,1)	0.001
Discharge, *n* (%)	66 (95.6)	191 (78.9)	0.001
CT lung involvement, %, Me + IQR [25; 75]	15 [5; 25]	48 [35; 70]	<0.001
CT grading, *n* (%)			
1 2 3	39 (56.5) 16 (23.2) 1 (1.4)	24 (9.9) 107 (44.2) 64 (26.4)	<0.001
BMI, kg/m^2^, Me + IQR [25; 75]	28 [25; 32]	30 [26; 33]	0.13
Obesity, *n* (%)	22 (31.8)	103 (42.6)	0.27
DM type 2, *n* (%)	22 (31.8)	82 (33.8)	0.70
AH, *n* (%)	48 (69.6)	192 (79.3)	0.08
IHD, *n* (%)	19 (27.5)	87 (35.9)	0.19
CKD, *n* (%)	5 (7.2)	29 (11.9)	0.26
25(OH)D, ng/mL, Me + IQR [25; 75]	36.1 [32.7; 38.0]	13.7 [8.2; 19.3]	<0.001
Neutrophils, ×10^9^/L, Me + IQR [25; 75]	4.8 [3.1; 6.5]	5.3 [3.4; 7.8]	0.12
Lymphocytes, ×10^9^/L, Me + IQR [25; 75]	1.2 [0.8; 1.5]	1.0 [0.8; 1.4]	0.08
NLR, Me + IQR [25; 75]	4.1 [2.4; 6.8]	5.2 [2.8; 9.1]	0.03
Plasma glucose, mmol/L, Me + IQR [25; 75]	6.6 [5.7; 8.2]	7.0 [6.1; 8.6]	0.04
CRP, mg/L, Me + IQR [25; 75]	28.3 [7.6; 62.5]	79.7 [32.5; 146.7]	<0.001
Ferritin, ng/mL, Me + IQR [25; 75]	411 [183; 786]	627 [289; 1155]	0.002
LDH, u/L, Me + IQR [25; 75]	331 [239; 455]	449 [316; 609]	<0.001
D-dimer, µg/mL FEU, Me + IQR [25; 75]	0.3 [0.2; 0.4]	0.4 [0.3; 0.7]	0.002
Creatinine, µmol/L, Me + IQR [25; 75]	71 [60.5; 82.5]	65 [55; 79]	0.043

ICU—intensive care unit; BMI—body mass index; DM—diabetes mellitus; AH—arterial hypertension; IHD—ischemic heart disease; CKD—chronic kidney disease; NLR—neutrophils/lymphocytes ratio; 25(OH)D—25-hydroxyvitamin D; CRP—C-reactive protein; LDH—lactate dehydrogenase; Me—median; and IQR—interquartile range.

**Table 3 pharmaceuticals-15-00305-t003:** Patients’ baseline characteristics depending on serum 25(OH)D levels.

Parameters	Serum 25(OH)D Level, ng/mL			*p*
	≥30.0 (*n* = 69)	20.0–29.9 (*n* = 57)	<20 (*n* = 185)	
Age, years, Me + IQR (25; 75)	58 [48; 69]	65 [55; 71]	64 [56; 72]	0.003
Gender, female, *n* (%)	37 (53.6)	29 (50.8)	91 (49.1)	0.38
Disease severity, *n* (%)				
Mild Moderate Severe	7 (10.2) 56 (81.1) 6 (8.7)	3 (1.3) 28 (49.1) 29 (50.8)	- 89 (48.1) 93 (50.2)	<0.001
Days from symptom onset to hospitalization, days, Me + IQR [25; 75]	7 [4; 10]	9 [6; 23]	8 [7; 11]	0.006
Bed days, days, Me + IQR [25; 75]	14 [9; 19]	16 [12; 22]	17 [13; 23]	0.002
ICU, *n* (%)	9 (13)	26 (45.6)	71 (38.5)	0.01
Discharge, *n* (%)	66 (95.6)	45 (78.9)	146 (78.9)	0.02
CT lung involvement, %, Me + IQR [25; 75]	15 [5; 25]	46.5 [35; 65]	50 [35; 70]	<0.001
BMI, kg/m^2^, Me + IQR [25; 75]	28 [25; 32]	28 [26; 32]	30 [26; 33]	0.26
Obesity, *n* (%)	22 (31.8)	21 (36.8)	82 (44.3)	0.18
DM type 2, *n* (%)	22 (31.8)	16 (28)	66 (35.6)	0.32
AH, *n* (%)	48 (69.6)	44 (77.2)	148 (80)	0.15
IHD, *n* (%)	19 (27.5)	16 (28)	71 (38.4)	0.09
CKD, *n* (%)	5 (7.2)	6 (10.5)	23 (12.4)	0.22
25(OH)D, ng/mL, Me + IQR [25; 75]	36.1 [32.7; 38.0]	25.4 [22.1; 26.6]	11.2 [7.2; 14.9]	<0.001
Neutrophils, ×10^9^/L, Me + IQR [25; 75]	4.8 [3.1; 6.5]	5.7 [3.1; 7.5]	5.2 [3.4; 7.8]	0.46
Lymphocytes, ×10^9^/L, Me + IQR [25; 75]	1.2 [0.8; 1.5]	1.1 [0.8; 1.5]	1.0 [0.7; 1.39]	0.02
NLR, Me + IQR [25; 75]	4.1 [2.4; 6.8]	4.8 [2.3; 9.1]	5.2 [2.7; 8.7]	0.006
Plasma glucose, mmol/L, Me + IQR [25; 75]	6.6 [5.7; 8.2]	6.7 [6.0; 8.2]	7.1 [6.2; 8.8]	0.038
CRP, mg/L, Me + IQR [25; 75]	28.3 [7.6; 62.5]	86.8 [31.8; 183.6]	77.9 [33.3; 132.6]	<0.001
Ferritin, ng/mL, Me + IQR [25; 75]	411 [183; 786]	594 [290; 836]	662 [289; 1193]	0.003
LDH, u/L, Me + IQR [25; 75]	331 [239; 455]	417 [318; 575]	430 [314; 628]	<0.001
D-dimer, µg/mL FEU, Me + IQR [25; 75]	0.3 [0.2; 0.4]	0.5 [0.3; 0.8]	0.4 [0.5; 0.7]	0.001
Creatinine, µmol/L, Me + IQR [25; 75]	71 [60.5; 82.5]	65 [55; 84]	67 [55; 79]	0.29

ICU—intensive care unit; BMI—body mass index; DM—diabetes mellitus; AH—arterial hypertension; IHD—ischemic heart disease; CKD—chronic kidney disease; NLR—neutrophils/lymphocytes ratio; 25(OH)D—25-hydroxyvitamin D; CRP—C-reactive protein; LDH—lactate dehydrogenase; Me—median; and IQR—interquartile range.

## Data Availability

The data generated and analyzed during this study are included in this published article. Additional information is available from the corresponding author on reasonable request.

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
