# Peer review of "Vitamin D Status and Immune Response in Hospitalized Patients with Moderate and Severe COVID-19"

_pharmaceuticals, 2022, doi:10.3390/ph15030305_

Round 1
Reviewer 1 Report
In this manuscript, the authors tried to find a connection between the concentration of Vitamin D in blood and the immune response in hospitalized patients who have moderate to severe COVID-19 infections.
Unfortunately, although the topic is very interesting, it is not in the area of my major specialty.
I recommend sending the manuscript to a colleague who is specialized in clinical medicine.
However, I found and corrected several grammatical mistakes (see the attached file).

Author Response
Dear Reviewer!
We would like to thank you for the review of the manuscript entitled “Vitamin D status and immune response in hospitalized patients with moderate and severe COVID-19” intended for publication in the Pharmaceuticals as an original research article.
We understand that the topic is not in the area of your major specialty, and still you reviewed and inserted corrections for grammatical mistakes which we appreciate.
Reviewer 2 Report
Dear Authors,
I have read the manuscript and I send you my comments:
1) Methods: please add the Ethic committe authorizatin and power calculation.
2) Methods: Regarding lalboratory tests, the Gold standard for vitamin D evaluation is the LC/MS, so add these value or cite the manuscript doi: 10.3390/jcm8060895, that documented a similar activity for CLIA and LC/MS
3) Discussion: a recent paper (doi: 10.3390/nu13113932) documented the role of vitamin D in COVID19, please cite and discuss it
Author Response
Dear Reviewer!
We would like to thank you for the careful consideration of the manuscript entitled “Vitamin D status and immune response in hospitalized patients with moderate and severe COVID-19” intended for publication in the Pharmaceuticals as an original research article.
Please find the revised manuscript and our responses to the comments. The changes made to the text have been highlighted.
1) Methods: please add the Ethic committee authorizatin and power calculation.
Answer: The information about Ethics Committee authorization is provided in the section entitled Institutional Review Board Statement – Page 16
The sample size consisted of 311 patients who met the inclusion criteria. Out of those 79 patients were randomly selected for further investigation of immunological markers.
2) Methods: Regarding laboratory tests, the Gold standard for vitamin D evaluation is the LC/MS, so add these value or cite the manuscript doi: 10.3390/jcm8060895, that documented a similar activity for CLIA and LC/MS
Answer: We understand the suggestion and hence we added information about confidential interval of this method in our Materials and methods section.
3) Discussion: a recent paper (doi: 10.3390/nu13113932) documented the role of vitamin D in COVID19, please cite and discuss it
Answer: As recommended we added information regarding this paper.

Reviewer 3 Report
Current report investigated the association between serum 25(OH)D level, immune response and disease clinical course in patients with COVID-19. Please conduct the concerns below.
- Language needs to check via a professional editing in advance.
- Role of Vitamin D in immunology has been introduced in clear. However, why applied the 25(OH)D mortality level (11.4 ng/mL) as point for separation in patients that needs to introduce in detail.
- In Table 1, the meaning of each value such as Plasma glucose (mmol/L) 6.5 (5.9;8.0) in Moderate Group and 7.7 (6.5;9.3) in Severe Group remained unclear. Please indicate it in the legends.
- Data shown that patients with COVID-19 stayed in ICU were longer in parallel with the decreased level of serum 25(OH)D level. Same changes were also observed in immune cells using the baseline data of patients with COVID-19. Is it enough to conclude the role of 25(OH)D? Please discuss it in detail.
- A higher frequency of Th2 and reduced frequency of Th17 and Tfh cells in blood samples were observed in COVID-19 patients with a serum 25(OH)D level > 11.4 ng/cc. However, only a decrease in the frequency of Tfh cells in patients with serum 25(OH)D level ≤ 11.4 ng/cc. Please describe the potential reason(s) for this truth.
- Is it suitable to suggest the baseline value of 25(OH)D at 11.4 ng/mL as the diagnostic indicator for progress of COVID-19 in clinic?
- Limitation of current study seems not enough.
Author Response
Dear Reviewer!
We would like to thank you for the careful consideration of the manuscript entitled “Vitamin D status and immune response in hospitalized patients with moderate and severe COVID-19” intended for publication in the Pharmaceuticals as an original research article.
Please find the revised manuscript and our responses to the comments. The changes made to the text have been highlighted.
- Language needs to check via a professional editing in advance.
Answer: The text language was additionally reviewed and updates inserted
- Role of Vitamin D in immunology has been introduced in clear. However, why applied the 25(OH)D mortality level (11.4 ng/mL) as point for separation in patients that needs to introduce in detail.
Answer: In our research we received data suggesting that a 25(OH)D level threshold of 11.4 ng/ml was associated with higher risks for mortality (AUC area = 0.811; sensitivity, 76%; and specificity, 77%; p=0.003). Hence with this information in mind were divided our 79 patients with COVID-19 into two groups: the first group consisted of patients with serum 25(OH)D level ≤ 11.4 ng/mL (n=20)), the second group – with serum 25(OH)D level > 11.4 ng/mL (n=59) for the assessment of the immunological markers.
- In Table 1, the meaning of each value such as Plasma glucose (mmol/L) 6.5 (5.9;8.0) in Moderate Group and 7.7 (6.5;9.3) in Severe Group remained unclear. Please indicate it in the legends.
Answer: We updated the Tables accordingly
- Data shown that patients with COVID-19 stayed in ICU were longer in parallel with the decreased level of serum 25(OH)D level. Same changes were also observed in immune cells using the baseline data of patients with COVID-19. Is it enough to conclude the role of 25(OH)D? Please discuss it in detail.
Answer: We identified changes in both vitamin D levels and impaired immune response. However, more research is needed to show the role of vitamin D.
- A higher frequency of Th2 and reduced frequency of Th17 and Tfh cells in blood samples were observed in COVID-19 patients with a serum 25(OH)D level > 11.4 ng/cc. However, only a decrease in the frequency of Tfh cells in patients with serum 25(OH)D level ≤ 11.4 ng/cc. Please describe the potential reason(s) for this truth.
Answer: We added the following text to our “Discussion” section: Circulating CD4+ T cells possess a range of helper and effector functions, and they are important for the control of different types of innate and adaptive immunity effector mechanisms as well as clearance of almost all viral infections including SARS-CoV-2. Furthermore, prompt and effective induction of SARS-CoV-2-specific CD4+ T cells and their presence in circulation within 2-4 day post-symptom onset in acute COVID-19 were associated with mild/moderate COVID-19 and good prognosis. Controversially, rapid induction of humoral responses and prolonged absence of SARS-CoV-2-specific CD4+ T cells were linked with an increase in disease severity and poor COVID-19 outcome. Thus, we can speculate that the presence of dramatic changes in Th cell subsets proportions could reflect the presence of effective immune response that was linked with the control and resolution of SARS-CoV-2 infection. Controversially, the absence of any reported alterations in Th cell subsets with the exception of Tfh cells could reflect the ineffective immunity to infection as well as was strongly associated with a high severity and a risk of fatal COVID-19.
- Is it suitable to suggest the baseline value of 25(OH)D at 11.4 ng/mL as the diagnostic indicator for progress of COVID-19 in clinic?
Answer: As we indicated in the text, we consider that further studies in this direction are needed to provide a more clear assessment
- Limitation of current study seems not enough.
Answer: Thank you for your comments, but we decided to delete the limitation on this paper because added information in the text

Reviewer 4 Report
- Lines 15-16: “For patients with vitamin D deficiency/…………… in those with normal level”, kindly mention the values of normal and low vitamin D in blood.
- Kindly, add the originality of your work as there were many published papers with the same idea such as:
- Szeto, B., Zucker, J. E., LaSota, E. D., Rubin, M. R., Walker, M. D., Yin, M. T., & Cohen, A. (2021). Vitamin D status and COVID-19 clinical outcomes in hospitalized patients. Endocrine research, 46(2), 66-73.
- The authors could add “immune response” to keywords.
- The authors would discuss the mechanism by which vitamin D deficiency affects COVID-19.
- Lines 48-49: “many tissues and cells, ……….vitamin D regulates a lot of genes” explain in more details with examples.
- Lines 55-56: “Normal 25(OH)D level ---------------nfection conditions” what is the normal level?
- “Fifty-six percent had moderate disease severity and 41% had severe disease” how did you classify the disease severity?
- Line 292, “In vitro” should be in Italic.
- The conclusion should be more comprehensive.
- “line 408: “O Analysis”, please correct
- Reference 12, is incomplete, and please double check the other references.
- The graphical abstract is highly recommended.
- Avoid using abbreviations without identifying them for the first time. For instance: 25(OH)D and CD3+CD4+Th cells.
- The authors could benefit from the following reference: “Khalifa, S. A., Swilam, M. M., El-Wahed, A. A. A., Du, M., El-Seedi, H. H., Kai, G., ... & El-Seedi, H. R. (2021). Beyond the Pandemic: COVID-19 Pandemic Changed the Face of Life. International Journal of Environmental Research and Public Health, 18(11), 5645.”
Author Response
Dear Reviewer!
We would like to thank you for the careful consideration of the manuscript entitled “Vitamin D status and immune response in hospitalized patients with moderate and severe COVID-19” intended for publication in the Pharmaceuticals as an original research article.
Please find the revised manuscript and our responses to the comments. The changes made to the text have been highlighted.
- Lines 15-16: “For patients with vitamin D deficiency/…………… in those with normal level”, kindly mention the values of normal and low vitamin D in blood.
Answer: Please note that information regarding the normal and low values of Vit D is provided in the section entitled “Materials and Methods”
- Kindly, add the originality of your work as there were many published papers with the same idea such as: Szeto, B., Zucker, J. E., LaSota, E. D., Rubin, M. R., Walker, M. D., Yin, M. T., & Cohen, A. (2021). Vitamin D status and COVID-19 clinical outcomes in hospitalized patients. Endocrine research, 46(2), 66-73.
Answer: We understand the global interest in similar topics, though the emphasis in this study is more on the immunologic aspect than the clinical outcomes.
- The authors could add “immune response” to keywords.
Answer: We updated accordingly
- The authors would discuss the mechanism by which vitamin D deficiency affects COVID-19.
Answer: Updates were made to include this
- Lines 48-49: “many tissues and cells, ……….vitamin D regulates a lot of genes” explain in more details with examples.
Answer: We corrected this offer
- Lines 55-56: “Normal 25(OH)D level ---------------nfection conditions” what is the normal level?
Answer: Please note that information regarding the normal and low values of Vit D is provided in the section entitled “Materials and Methods”
- “Fifty-six percent had moderate disease severity and 41% had severe disease” how did you classify the disease severity?
Answer: this information was added
- Line 292, “In vitro” should be in Italic.
Answer: This was corrected
- The conclusion should be more comprehensive.
Answer: We added the information in our “Conclusion”
- “line 408: “O Analysis”, please correct
Answer: This was corrected
- Reference 12, is incomplete, and please double check the other references.
Answer: Additional check of the reference was made
- The graphical abstract is highly recommended.
Answer: We appreciate the recommendation, however we will refrain from including this, but we added study design
- Avoid using abbreviations without identifying them for the first time. For instance: 25(OH)D and CD3+CD4+Th cells.
Answer: We carefully checked the text and identified all used abbreviations.
List of added new abbreviations:25-hydroxycholecalciferol - 25(OH)D; natural killers - NK cells; interleukin – IL; T-helpers – Th; C-reactive protein – CRP; lactate dehydrogenase – LDH; Neutrophils/ Lymphocytes ratio – NLR; computer tomography – CT; intensive care unit – ICU; central memory – CM; effector memory – EM; T effector memory re-expressing CD45RA – TEMRA; double-positive – DP; polymerase chain reaction - PCR
- The authors could benefit from the following reference: “Khalifa, S. A., Swilam, M. M., El-Wahed, A. A. A., Du, M., El-Seedi, H. H., Kai, G., ... & El-Seedi, H. R. (2021). Beyond the Pandemic: COVID-19 Pandemic Changed the Face of Life. International Journal of Environmental Research and Public Health, 18(11), 5645.”
Answer: That you for the suggestion, we added this as a reference

Round 2
Reviewer 2 Report
Dear Authors thank you for your revisions
Reviewer 3 Report
It has been revised mainly following the comments.
Reviewer 4 Report
Accept in present form